# Coherence distillation machines are impossible in quantum thermodynamics

Iman Marvian[1]*

The role of coherence in quantum thermodynamics has been extensively studied in the recent years and it is now well-understood that coherence between different energy eigenstates is a resource independent of other thermodynamics resources, such as work. A fundamental remaining open question is whether the laws of quantum mechanics and thermodynamics allow the existence of a coherence distillation machine, i.e., a machine that, by possibly consuming work, obtains pure coherent states from mixed states, at a nonzero rate. This is related to another fundamental question: Starting from many copies of noisy quantum clocks which are (approximately) synchronized with a reference clock, can one distill synchronized clocks in pure states, at a non-zero rate? Surprisingly, we find that the answer to both questions is negative for generic (full-rank) mixed states. However, at the same time, it is possible to distill a sub-linear number of pure coherent states with a vanishing error.

[1] Departments of Physics & Electrical and Computer Engineering, Duke University, Durham, NC 27708, USA. *email: iman.marvian@duke.edu

What are the fundamental limits of nature on manipulation of quantum clocks? Suppose we have multiple clocks, all synchronized with the same reference clock, which are affected by noise. Then, by averaging the time read from these clocks we can obtain a more accurate estimate of the current time according to the reference clock. In other words, we can distill a less noisy clock from several noisy clocks. What are the limits of this distillation process for quantum clocks? Can we distill quantum clocks in pure states from those in mixed states, at a nonzero rate?

Interestingly, this question is related to another fundamental question about the manipulation of coherence in quantum thermodynamics. It is now well-understood that coherence between different energy eigenstates is a resource, independent of other thermodynamic resources such as work, and can be used to implement operations which are otherwise impossible[1–4]. A fundamental open question in this context is whether the laws of quantum mechanics and thermodynamics allow the existence a coherence distillation machine, i.e., a machine that consumes work to obtain pure coherent states from mixed ones at a nonzero rate (See Fig. 1). The connection between these two questions arises from the fact that the minimum requirement for a system to be a clock is to be in a state which contains coherence (i.e., off-diagonal terms) with respect to the energy-eigenbasis; otherwise, the system will be time-independent, and hence useless as a clock.

In this article, we investigate coherence distillation in the context of quantum thermodynamics, both in the single-shot and asymptotic regimes. In particular, we settle the above questions, which have been open heretofore[5,6], and show that the answer to both of them is negative. In other words, the coherence distillation machine, depicted in Fig. 1, is impossible. This is surprising, especially when compared to the previously known results on resource distillation in the entanglement theory and other quantum resource theories (See e.g.,[7–11]), and reveals important aspects of coherence in quantum thermodynamics. In particular,

we will see that, in some precise sense, the coherence content of a single two-level system can be infinitely large. Furthermore, we find that, even though distillation with a non-zero rate is impossible, it is still possible to distill a sublinear number of pure coherent states with a vanishing error. We also consider coherence distillation in the single-shot regime and derive a simple formula for the maximum achievable fidelity.

## Results

**Distillation of quantum clocks.** A quantum clock is characterized by its state and Hamiltonian, which usually generates a periodic time evolution[12–18]. By definition, the state of a clock should be time-dependent. Therefore, when we say a clock with Hamiltonian $H$ is in state $\rho$, we actually mean its state is $\rho$ at a particular time, say $t = 0$, with respect to a reference clock. Then, at an arbitrary time $t$ the state of clock is $e^{-iHt}\rho e^{iHt}$ (Throughout this paper we assume $\hbar = 1$). Here, we focus on the systems with bounded Hamiltonians, with periodic dynamics, whose period is equal to a fixed (but arbitrary) parameter $\tau$, such that $\tau = \min\{t > 0 : e^{-iHt}\rho e^{iHt} = \rho\}$; otherwise, the state and Hamiltonian are completely arbitrary. In the following, when we talk about $n$ copies of a system with state $\rho$ and Hamiltonian $H$, we mean $n$ non-interacting systems, with the total Hamiltonian $\sum_{i=1}^{n} H^{(i)}$, where $H^{(i)} = I^{\otimes(i-1)} \otimes H \otimes I^{\otimes(n-i-1)}$, and with the joint state $\rho^{\otimes n}$.

Suppose Alice is given a quantum clock with Hamiltonian $H_{in}$ and state $\rho_{in}$, synchronized with a standard reference clock owned by Bob. Assume she does not have any additional information about Bob's clock. In other words, she knows at time $t$ relative to Bob's clock, her quantum clock is in state $e^{-iH_{in}t}\rho_{in}e^{iH_{in}t}$; however, the parameter $t$ itself is unknown to her.

Now suppose Alice wants to transform this clock to a different clock, with possibly different Hamiltonian $H_{out}$, which is still synchronized with Bob's clock, such that at any time $t$ relative to his clock the new quantum clock is in state $e^{-iH_{out}t}\rho_{out}e^{iH_{out}t}$. For instance, the input clock with Hamiltonian $H_{in}$ can be multiple copies of a noisy two-level clock in a mixed state, whereas the output clock is a single two-level system, which is more accurate than any single copy at the input, i.e., conveys more information about the parameter $t$ (This is an example of single-copy distillation of clocks, which will be discussed later). This means that Alice wants to implement the state conversion

$$e^{-iH_{in}t}\rho_{in}e^{iH_{in}t} \rightarrow e^{-iH_{out}t}\rho_{out}e^{iH_{out}t}, \ \forall t \in [0, \tau) . \quad (1)$$

However, since parameter $t$ is unknown to her, this conversion should be implemented by a fixed process, independent of $t$; i.e., there should exist a physical process, described by a completely positive trace-preserving[19,20] map $\mathcal{E}$, such that $\mathcal{E}(e^{-iH_{in}t}\rho_{in}e^{iH_{in}t}) = e^{-iH_{out}t}\rho_{out}e^{iH_{out}t}$, for all time $t \in [0, \tau)$. It turns out that this is possible if, and only if, the single state conversion $\rho_{in} \rightarrow \rho_{out}$ is possible under a Time-translation Invariant (TI) process, i.e., a process satisfying the covariance condition

$$e^{-iH_{out}s}\mathcal{E}_{TI}(\sigma)e^{iH_{out}s} = \mathcal{E}_{TI}(e^{-iH_{in}s}\sigma e^{iH_{in}s}), \quad (2)$$

for all times $s$, and input $\sigma$[21,22]. Therefore, rather than studying the state conversions for the family of states in Eq. (1), one can equivalently study state conversion for the single input-output pair $\rho_{in}$ and $\rho_{out}$ under the restricted set of TI operations.

The covariance condition in Eq. (2) means that TI processes are those which can be defined, and hence implemented, independent of any reference clock. Furthermore, they can be implemented without interfering with the intrinsic time evolution generated by the system Hamiltonian. An example of this type of processes is energy-conserving unitary transformations, i.e., those which commute with the Hamiltonian (assuming the input and

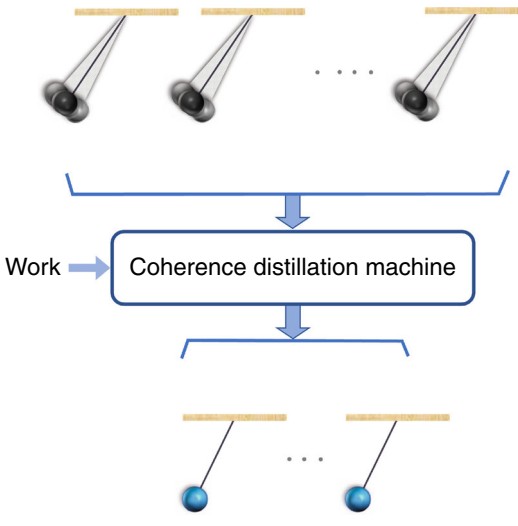

**Fig. 1 A hypothetical "Coherence Distillation Machine" for distilling coherence with respect to the energy eigenbasis.** It consumes work and obtains pure coherent states from mixed states at a non-zero rate, or equivalently, purifies quantum clocks. Is this hypothetical machine consistent with the laws of quantum mechanics and thermodynamics?.

Work ➔ Coherence distillation machine

output systems have identical Hamiltonians). There are also TI operations which are not energy-conserving, such as, preparing the system in an incoherent state, i.e., any state $\rho$ commuting with the system Hamiltonian (Note that in the case of composite systems, the joint state is incoherent if it commutes with the total Hamiltonian).

In summary, we conclude that for distillation or manipulation of quantum clocks, we can restrict our attention to the set of TI operations. In the language of quantum resource theories[7,23–26], these are the free operations for the resource theory of quantum clocks, which is a special case of the resource theory of asymmetry.

It is worth emphasizing that the notion of resource distillation, which can be abstractly defined in any resource theory, has a clear operational interpretation in this framework: it is the process in which one combines noisy clocks, affected by independent noise processes, to obtain less, but more accurate clocks in pure states. More precisely, the information content of each output clock about the unknown parameter $t$, i.e., the current time relative to the standard clock, is greater than the information content of each input clock. Hence, using a distillation protocol, one can increase the efficiency of storage and transmission of quantum clocks. Intuitively, one expects that to maximize the information content about parameter $t$, the state of quantum clock should be pure. This intuition is confirmed by the fact that pure states maximize any convex measure of information (about the time parameter $t$) such as Holevo quantity[19,20,27] or quantum Fisher information[27–30]. Similarly, from the point of view of parameter estimation, to minimize the error in the estimation of the time parameter $t \in [0, \tau)$, as quantified by any cost function which is a linear functional of state, such as mean squared error[27,31], the system should be prepared in a pure state.

Interestingly, as we see next, the set of TI operations also naturally arises in the study of coherence in quantum thermodynamics. It is worth mentioning that, in this paper we focus on a notion of coherence which is relevant in the context of quantum clocks and quantum thermodynamics, known as unspeakable coherence[6,32]. This notion of coherence is a special case of a more general property, called asymmetry[32–35]. There are other resource theoretic approaches to coherence, capturing a different notion of coherence, known as speakable coherence[6,32] (In these resource theories the eigenvalues of the system Hamiltonian do not play any role).

**Coherence distillation machines.** A coherence distillation machine, as depicted in Fig. 1, receives systems in a mixed coherent state, and transforms them to pure coherent states, at a non-zero rate. Recall that a quantum state contains coherence, or is coherent, if its density operator does not commute with its Hamiltonian. In the following, we consider two different frameworks for describing coherence distillation machines and, interestingly, find that they are equivalent and both lead to the notion of TI operations.

Our first approach is to consider the most general processes which can be interpreted as "coherence distillation machines". What are the constraints on such operations? Clearly, a distillation machine should not generate coherence itself, i.e., should transform incoherent states to incoherent states; otherwise, the coherence at the output cannot be interpreted as distilled coherence. This should hold even if the input is entangled with another closed system with an arbitrary Hamiltonian; if their initial joint state commutes with their total Hamiltonian, then their final state should also commute, and hence be incoherent (See Fig. 2).

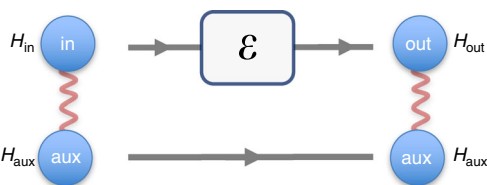

**Fig. 2 Completely incoherence-preserving operations.** Suppose the joint initial state of the input system and an auxiliary system with Hamiltonian $H_{\text{aux}}$ is incoherent with respect to their total Hamiltonian $H_{\text{in}} \otimes I_{\text{aux}} + I_{\text{in}} \otimes H_{\text{aux}}$. Quantum operation $\mathcal{E}$ is called completely incoherence-preserving if for any choices of $H_{\text{aux}}$ and the initial incoherent state, the joint state of the output and the auxiliary system is also incoherent with respect to their total Hamiltonian $H_{\text{out}} \otimes I_{\text{aux}} + I_{\text{out}} \otimes H_{\text{aux}}$. We show that any such operation is a TI operation and can be implemented by coupling the system to a work reservoir (battery) by an energy-conserving unitary.

We prove that a quantum operation satisfies this property, or is completely incoherence-preserving, iff it is a TI operation (See Supplementary Note 1). This means that, by proving the impossibility of coherence distillation using TI operations, we also establish its impossibility under completely incoherence-preserving operations, which describe the most general processes relevant to coherence distillation.

A different approach to formalizing coherence distillation is to use the framework of the resource theory of quantum thermodynamics (athermality) and the notion of thermal operations[24,26,36–40]. Thermal operations are those which can be implemented by coupling the system to a thermal bath by energy-conserving unitaries. It turns out that under these operations coherence and work are two independent resources[1,2]. Therefore, to focus on coherence, one can supplement a thermal operation with an unlimited amount of work at the input (using a battery or work reservoir), which can be modeled as an auxiliary system in an energy eigenstate. What is the set of all operations which can be implemented in this way? Interestingly, it turns out that the answer is again TI operations. In particular, any TI operation $\mathcal{E}_{\text{TI}}$ on a system $S$ with Hamiltonian $H_S$ can be implemented by coupling the system to an auxiliary system (battery) with Hamiltonian $H_{\text{bat}}$, such that

$$\mathcal{E}_{\text{TI}}(\sigma) = \text{Tr}_{\text{bat}} U(\sigma \otimes |E\rangle\langle E|_{\text{bat}})U^{\dagger}, \qquad (3)$$

where (i) the initial state $|E\rangle_{\text{bat}}$ of the auxiliary system is an eigenstate of its Hamiltonian $H_{\text{bat}}$, and (ii) the unitary $U$ that couples it to the system $S$ conserves the total energy $H_{\text{tot}} = H_S \otimes I_{\text{bat}} + I_S \otimes H_{\text{bat}}$, i.e., $[U, H_{\text{tot}}] = 0$ (See Supplementary Note 1, ref. [41], and theorem 25 of ref. [22]).

We conclude that formalizing the notion of coherence distillation machines in the framework of the resource theory of quantum thermodynamics (athermality), again leads us to the notion of TI operations.

To summarize, we saw three different properties, each of which can characterize exactly the same set of operations, namely TI operations: (**a**) invariance under time-translations, (**b**) being completely incoherence-preserving, and (**c**) being implementable with thermal operations supplemented with an arbitrary amount of work. Next, we study distillation of coherence using these processes.

**Main theorem: Typical states have no distillable coherence.** An ideal coherence distillation machine is a TI operation (or, equivalently, a completely incoherence-preserving operation) which consumes copies of a system in a mixed state $\rho$ as the

resource, to generate copies of a system in a pure coherent state $\phi_{\mathrm{coh}}$, at rate $R > 0$, i.e., $\rho^{\otimes n} \to^{\mathrm{TI}} \phi_{\mathrm{coh}}^{\otimes \lceil Rn \rceil}$. Note that, in general, the Hamiltonians and the Hilbert spaces of the input and output systems can be different. Also, note that $\phi_{\mathrm{coh}}$ can be any pure state of the output system, except the energy eigenstates (For instance, one can choose a two-level system with Hamiltonian $\pi \sigma_z / \tau$, and state $|\phi_{\mathrm{coh}}\rangle = (|0\rangle + |1\rangle)/\sqrt{2}$, where $\tau$ is the period).

In practice, exact transformations are often impossible and physically intractable. Therefore, we can allow a small error $\epsilon$ in infidelity[20], provided that it vanishes in the limit of infinite copies, i.e., $\rho^{\otimes n} \to^{\mathrm{TI}} \overset{\epsilon}{\approx} \phi_{\mathrm{coh}}^{\otimes \lceil Rn \rceil}$ as $n \to \infty, \epsilon \to 0$ (Recall that infidelity is one minus fidelity, i.e., $1 - \langle \psi | \sigma | \psi \rangle$ for state $\sigma$ and a pure state $\psi$. Infidelity is closely related to the trace distance[20]). Then, by the Helstrom's theorem[20,28], in the limit $n \to \infty$, the actual output state is indistinguishable from the desired state $\phi_{\mathrm{coh}}^{\otimes \lceil Rn \rceil}$.

Consider an arbitrary system with bounded Hamiltonian $H$ and state $\rho$. The distillable coherence $C_{\mathrm{d}}^{\mathrm{TI}}(\rho)$, relative to any standard pure coherent state $\phi_{\mathrm{coh}}$, is the maximum rate at which copies of $\phi_{\mathrm{coh}}$ can be obtained from copies of this system using TI operations (or, equivalently, using completely incoherence-preserving operations),

$$C_{\mathrm{d}}^{\mathrm{TI}}(\rho) \equiv \sup \; R : \rho^{\otimes n} \to^{\mathrm{TI}} \overset{\epsilon}{\approx} \phi_{\mathrm{coh}}^{\otimes \lceil Rn \rceil} \text{ as } n \to \infty, \epsilon \to 0, \quad (4)$$

where the error $\epsilon$ is vanishing in infidelity (one minus fidelity). Note that this definition resembles the definition of the distillable entanglement[9–11,42,43], or, more generally, distillable resource in any resource theory (See e.g.,[7,8]). We prove the following fundamental no-go theorem on coherence distillation:

*Theorem.* If the projector to the support of state $\rho$ commutes with the system Hamiltonian $H$, then the rate of distillation of any system in a pure coherent state $\phi_{\mathrm{coh}}$ is zero, i.e., $C_{\mathrm{d}}^{\mathrm{TI}}(\rho) = 0$. Thus, for a typical state $\rho$, which has full-rank density operator, this rate is zero.

Surprisingly, we find that the hypothetical coherence distillation machine depicted in Fig. 1 is impossible, i.e., starting from asymptotically many copies of a generic mixed state, using a thermal machine we cannot distill pure coherence at a nonzero rate, even if we spend an unlimited amount of work. In fact, it turns out that coherence distillation remains impossible even if, in addition to copies of state $\rho$, one is allowed to consume a finite helper system in a pure state, provided that its Hamiltonian is bounded and its Hilbert space is finite-dimensional (See Supplementary Note 5). It is interesting to compare this result with the results of[5] and[8], which prove that the rate of distillation of speakable coherence is generally non-zero.

Finally, it is worth mentioning that although for a typical mixed state the distillable coherence is zero, there are also mixed states with non-zero distillable coherence. The problem of classifying all such states, and determining the optimal rate of conversion remains open. In Supplementary Note 6 we present examples of such states, and find an achievable distillation rate, which is closely related to a Petz-Rényi relative entropy. These examples rely on the previously known results on state conversions between pure states[33,44–46], which show that the optimal rate of conversion from a system with the pure state $\psi_1$ and Hamiltonian $H_1$ to another system with the pure state $\psi_2$ and Hamiltonian $H_2$, provided that they have the same period, is $R = V_{H_1}(\psi_1)/V_{H_2}(\psi_2)$, where $V_H(\psi) \equiv \langle \psi | H^2 | \psi \rangle - \langle \psi | H | \psi \rangle^2$ is the energy variance for state $\psi$.

Next, we explain how the above no-go theorem follows from an interesting relation between two quantifiers of coherence, namely quantum Fisher information and a new quantifier, called the purity of coherence.

**Purity of coherence**. In recent years, many quantifiers of coherence and asymmetry have been studied (See, for instance,[22,34,47]). These previously known examples, however, all fail to see a simple, yet fundamental feature of coherence: Given any finite copies of a generic mixed state, it is impossible to generate a single copy of a pure coherent state (with a non-zero probability), using only TI operations. Here, we introduce a new quantifier of coherence which captures the missing part of the picture and predicts the unreachability of pure coherent states.

For a system with state $\rho$, let the Purity of Coherence with respect to the eigenbasis of an observable $H$ be

$$P_H(\rho) \equiv \mathrm{Tr}(H \rho^2 H \rho^{-1}) - \mathrm{Tr}(\rho H^2) \quad (5)$$

$$= \sum_{j,k} \frac{p_k^2 - p_j^2}{p_j} |\langle \psi_k | H | \psi_j \rangle|^2 , \quad (6)$$

if $\mathrm{supp}(H \rho H) \subseteq \mathrm{supp}(\rho)$, and $P_H(\rho) = \infty$ otherwise, where $\rho = \sum_j p_j |\psi_j\rangle \langle \psi_j|$ is the spectral decomposition of $\rho$.

As we discuss below, this function is an example of a generalized family of Fisher information introduced by Petz[53,54]. Also, in Supplementary Note 2 we show that this function can be thought as the second derivative of Petz-Rényi relative entropy (for $\alpha = 2$)[55,56]. Using this fact we show that purity of coherence is (i) non-negative and it becomes zero iff state is incoherent, (ii) non-increasing under any TI operation $\mathcal{E}_{\mathrm{TI}}$, i.e., $P_{H_{\mathrm{out}}}(\mathcal{E}_{\mathrm{TI}}(\rho)) \leq P_{H_{\mathrm{in}}}(\rho)$. In particular, it is invariant under energy-conserving unitaries. (iii) Additive: for uncorrelated composite systems which are not interacting with each other, i.e., $P_{H_{\mathrm{tot}}}(\rho_1 \otimes \rho_2) = P_{H_1}(\rho_1) + P_{H_2}(\rho_2)$, where $H_{\mathrm{tot}} = H_1 \otimes I_2 + I_1 \otimes H_2$, and (iv) a convex function of $\rho$.

The above definition implies that for pure states the purity of coherence is $\infty$, unless the state is an energy eigenstate, in which case it is zero. This unboundedness of the purity of coherence, captures the unreachability of pure coherent states from generic mixed states: Suppose there exists a TI operation which receives $n$ copies of a system with state $\rho_1$ and Hamiltonian $H_1$, and with probability of success $p$, transforms them to a single copy of a system with state $\rho_2$ and Hamiltonian $H_2$. Using properties (i-iv), in Supplementary Note 2 we show

$$n \geq p \times \frac{P_{H_2}(\rho_2)}{P_{H_1}(\rho_1)} . \quad (7)$$

Thus, to generate a single copy of a pure coherent state $\rho_2$, we need $n = \infty$ or $P_{H_1}(\rho_1) = \infty$. These properties of purity of coherence make it a powerful tool to study coherence distillation, both in the asymptotic and single-shot regimes.

**Relation with Quantum Fisher Information**. It turns out that the purity of coherence has an interesting relation with Quantum Fisher Information (QFI), and this relation plays a crucial role in the proof of our no-go theorem. Recall that for the family of states $\{e^{-iHt} \rho e^{iHt}\}_t$, QFI associated to the time parameter $t$ is

$$F_H(\rho) = 2 \sum_{j,k} \frac{(p_j - p_k)^2}{p_j + p_k} |\langle \psi_j | H | \psi_k \rangle|^2 . \quad (8)$$

where $\rho = \sum_j p_j |\psi_j\rangle \langle \psi_j|$ is the spectral decomposition of $\rho$. QFI is the central quantity of quantum metrology and estimation theory[27–30,54], and has found extensive applications in different areas of physics (See e.g.,[57–62]). QFI satisfies properties (i-iv) listed above for the purity of coherence. In particular, it is additive and monotone under TI operations.

A closer look at the properties of the purity of coherence and QFI reveals an interesting relation between them: First, comparing Eq. (6) and Eq. (8), one can easily show that the purity of coherence is always larger than or equal to QFI, i.e., $P_H(\rho) \geq F_H(\rho)$, and the equality holds iff $\rho$ is incoherent. Furthermore, for two-level systems, we find the nice formula

$$P_H(\rho) = \frac{F_H(\rho)}{2[1 - \mathrm{Tr}(\rho^2)]}, \qquad (9)$$

i.e., the purity of coherence is determined by a combination of QFI and the purity, $\mathrm{Tr}(\rho^2)$. This means that, for states close to the maximally mixed state, $P_H(\rho)/F_H(\rho) \approx 1$, whereas for states close to a generic pure state, $P_H(\rho)$ can be arbitrarily larger than $F_H(\rho)$. We show that these properties hold beyond two-level systems: In general, if $\rho$ is $\epsilon$-close to the maximally mixed state in infidelity, then $\frac{P_H(\rho)}{F_H(\rho)} = 1 + \mathcal{O}(\sqrt{\epsilon})$. In the opposite limit, where $\rho$ is close to a pure state, we find $P_H(\rho) \geq \frac{1}{4} F_H(\psi_{\max}) \times [\frac{p_{\max}^2}{1 - p_{\max}} - 1]$, where $p_{\max}$ is the largest eigenvalue of $\rho$, and $\psi_{\max}$ is the corresponding eigenvector (See Supplementary Note 3). Again, as $\rho$ converges to a pure state, the purity $\mathrm{Tr}(\rho^2)$ and $p_{\max}$ converge to one. In this case, $P_H(\rho)$ diverges, unless the pure state is an energy eigenstate.

We conclude that, roughly speaking, the purity of coherence $P_H(\rho)$ is lower bounded by the ratio of QFI (for a pure state close to $\rho$) to one minus the purity of state; hence, higher $P_H(\rho)$ means more pure coherence, which justifies its name.

It is interesting to note that the relation between the purity of coherence and QFI is analogous to the relation between the total and free energies in thermodynamics; the latter distinguishes ordered (low-entropy) energy and disordered (high-entropy) energy. Similarly, the purity of coherence, can recognize the distinction between the pure and mixed coherence. It turns out that for some operations, such as coherence distillation, the same amount of coherence quantified by QFI in states with more purity is a more useful resource.

**RLD and SLD Fisher information**. It is worth mentioning that both of these quantifiers of coherence, i.e., the purity of coherence $P_H$ and QFI $F_H$, are specials cases of a generalized family of Fisher Information. Classically, Fisher information is the unique (up to a normalization) stochastically monotone Riemannian metric on the space of probability distributions[63]. In the quantum case, on the other hand, there is a family of monotone metrics on the space of density operators, which is fully characterized by Petz[53,54] (See also ref. [63]). Interestingly, functions $P_H$ and $F_H$ are extremal points in this family: they are, respectively, the maximal and minimal monotone metrics calculated for the one-parameter family of states $\{e^{-iHt}\rho e^{iHt}\}_t$. In quantum estimation literature, these functions are often respectively called Right Logarithmic Derivative (RLD) and Symmetric Logarithmic Derivative (SLD) Fisher Information. Following the physics literature convention, here we have referred to SLD Fisher information as Quantum Fisher Information (QFI).

Remarkably, these two extremal functions have also distinguished roles in the resource theory of (unspeakable) coherence and quantum clocks: it has been recently shown that QFI (SLD Fisher Information) determines the coherence cost, i.e., the minimum rate of consumption of standard pure coherent states that is needed to generate the desired mixed state, using TI operations[46]. Also, it is well-known that QFI determines the lowest achievable mean square error for estimating the time parameter. On the other hand, it turns out that the purity of coherence (RLD Fisher Information) is relevant in the context of coherence distillation (See Fig. 3), and provides a powerful tool for proving our no-go theorem on coherence distillation.

**Proof of the main theorem**. To prove the impossibility of coherence distillation machines, we use the properties of the purity of coherence, namely its monotonicity and additivity, and its relation with QFI. Note that the impossibility of distillation cannot be shown using QFI alone, because it increases linearly in $n$, for both the input and the desired output states. As we explain in the following, the main challenge in proving this theorem is the fact that QFI and the purity of coherence are not asymptotically continuous[64].

In Supplementary Note 4 we prove the following result, which is of independent interest: Consider $m$ non-interacting systems, each with Hamiltonian $H$, and with the total Hamiltonian $H_{\mathrm{tot}} = \sum_{i=1}^{m} H^{(i)}$, in the joint state $\sigma_m$. Suppose the fidelity of $\sigma_m$ and state $|\phi\rangle^{\otimes m}$, is $\langle \phi|^{\otimes m} \sigma_m |\phi\rangle^{\otimes m} = 1 - \epsilon$. Then, for sufficiently large $m$, e.g., $m \geq 70 \frac{|\langle \phi|H^3|\phi\rangle|^2}{V_H^3(\phi)}$ and sufficiently small $\epsilon$, e.g., $\epsilon \leq 10^{-3}$, QFI and the purity of coherence of state $\sigma_m$ relative to the total Hamiltonian $H_{\mathrm{tot}}$, are lower bounded by

$$F_{H_{\mathrm{tot}}}(\sigma_m) \geq 4c \times m \times F_H(\phi), \qquad (10)$$

$$P_{H_{\mathrm{tot}}}(\sigma_m) \geq c \times m \times F_H(\phi) \times \frac{1}{\epsilon}, \qquad (11)$$

where $c$ is a positive constant, e.g., $c = 10^{-2}$ (Recall that for a pure state $\phi$, QFI is $F_H(\phi) = 4V_H(\phi)$). Note that similar to the case of a single qubit in Eq. (9), the lower bound on the purity of coherence in Eq. (11) grows linearly with $\epsilon^{-1}$.

At first glance, these bounds might seem intuitive from our previous discussions: For instance, Eq. (10) means that to be able to have a large fidelity with state $\phi^{\otimes m}$, QFI of state $\sigma_m$ should also grow (at least) linearly with $m$, which might be expected from the additivity of QFI. However, a more careful analysis is needed: the Hamiltonian $H_{\mathrm{tot}}$ has eigenvalues of order $m \times \| H \|$, which means relative to this Hamiltonian, two states with infidelity $\epsilon$ can have QFI's which differ by order $\epsilon \times m^2 \| H \|^2$. Thus, while one state can have a large QFI, e.g., linear in $m$, the other might have a negligible QFI. This makes the proof of the above bounds non-trivial.

Now suppose there exists a TI operation $\mathcal{E}_n$ which converts $\rho^{\otimes n}$ to state $\sigma_{m(n)}$ whose fidelity with the desired state $\phi_{\mathrm{coh}}^{\otimes m(n)}$ is $1 - \epsilon_n$. To simplify the notation, we assume the Hamiltonian of each copy at the input is the same as the Hamiltonian of each copy at the output, which is denoted by $H$ (This assumption is not needed for the proof). Then, using the additivity of the purity of coherence, the total purity of coherence of the input is $n \times P_H(\rho)$. Since this quantity is monotone under TI operations, the purity of coherence of the output is $P_{H_{\mathrm{tot}}}(\sigma_{m(n)}) \leq n \times P_H(\rho)$. Combined with Eq. (11), this leads to

$$\frac{m(n)}{n} \leq \frac{1}{c} \times \frac{P_H(\rho)}{F_H(\phi_{\mathrm{coh}})} \times \epsilon_n. \qquad (12)$$

This interesting inequality implies that to make error $\epsilon_n$ small, the yield $m(n)/n$ should also be small, unless $F_H(\phi_{\mathrm{coh}}) = 0$, i.e., $\phi_{\mathrm{coh}}$ is incoherent, or $P_H(\rho) = \infty$. Thus, if $P_H(\rho)$ is bounded and $\phi_{\mathrm{coh}}$ is coherent, then to have vanishing error $\epsilon_n \to 0$, we also need to have vanishing yield, $\lim_{n \to \infty} m(n)/n = 0$, which means the distillable coherence is zero. We show that for a bounded Hamiltonian $H$, $P_H(\rho) < \infty$ iff $\Pi_\rho$, the projector to the support of $\rho$, commutes with $H$. We conclude that if $[\Pi_\rho, H] = 0$, then the distillable coherence is zero, which proves the theorem.

**Sub-linear Coherence Distillation: Trade-off between the maximum achievable yield and fidelity**. Even though for states with finite purity of coherence the distillable coherence is zero,

interestingly, it turns out that any state which contains coherence can still be used to distill a sub-linear number of pure coherent states. In the above scenario, let $m_{\mathrm{opt}}(n)$ be the maximum number of copies of $\phi_{\mathrm{coh}}$ which can be distilled with error less than $\epsilon_n$, and $r_{\mathrm{opt}}(n) = m_{opt}(n)/n$ be the maximum achievable yield. Assuming the input and output systems have the same period, the ratio of $r_{\mathrm{opt}}(n)$ to error $\epsilon_n$ satisfies

$$4[1 - o(1)] \times \frac{F_H(\rho)}{F_H(\phi_{\mathrm{coh}})} \leq \frac{r_{\mathrm{opt}}(n)}{\epsilon_n} \leq \frac{1}{c} \times \frac{P_H(\rho)}{F_H(\phi_{\mathrm{coh}})}, \qquad (13)$$

where the upper bound on $r_{\mathrm{opt}}(n)/\epsilon_n$ follows from Eq. (12), and holds assuming the number of distilled copies is sufficiently large, e.g., $m_{opt}(n) \geq 70 \frac{|\langle\phi_{\mathrm{coh}}|H^3|\phi_{\mathrm{coh}}\rangle|^2}{V_H^3(\phi_{\mathrm{coh}})}$, and error $\epsilon_n$ is sufficiently small, e.g., $\epsilon_n \leq 10^{-3}$. These assumptions are not required for the lower bound.

This means that there is a trade-off between fidelity and yield. For instance, for sufficiently large $n$, one can achieve the yield $r(n) = 4\frac{F_H(\rho)}{F_H(\phi_{\mathrm{coh}})}n^{-\alpha}$, for arbitrary exponent $\alpha > 0$, with infidelity $\epsilon_n = n^{-(\alpha-\delta)}$ where $\delta > 0$ can be arbitrary small. Choosing smaller $\alpha > 0$, means higher yield and also larger error. This should be compared with the recent results on distillation of speakable coherence[65–68] (In particular, in the case of strictly incoherent operations[5,69], there are bound states, which cannot be converted to a single copy of a pure coherent state with a vanishing error, even if one is given an arbitrary many copies of state[66–68]). This tradeoff and the linear relation between the yield and error, which highlights the significance of yield-to-error ratio as a fundamental quantity, are unique features of this resource theory, which have practical implications in the context of quantum clocks, and are worth further study.

In the Methods section, we also discuss an interesting corollary of this result, namely a novel operational explanation of the violation of the monotonicity of Petz-Rényi relative entropy under data processing, for the parameter range $\alpha > 2$[55,56].

To establish the lower bound on $r_{\mathrm{opt}}(n)/\epsilon_n$ in Eq. (13), we consider a TI process defined based on a parameter estimation task: Suppose one is given $n$ copies of state $e^{-iHt}\rho e^{iHt}$, where $t \in [0, \tau]$ is unknown (Recall that $\tau$ is the period of both the input and the desired output systems). Measuring these systems, one can obtain an estimate $t_{\mathrm{est}} \in [0, \tau]$ of $t$, with probability density $p(t_{\mathrm{est}}|t)$. We can assume the estimator is invariant under time-translations, such that $p(t_{\mathrm{est}}|t) = p(t_{\mathrm{est}} - s|t - s) : \forall s \in [0, \tau]$, where the subtraction is mod $\tau$; if this is not the case, one can always make the estimator invariant by adding a random time translation to the input state, and then canceling it at the output of the estimator (See Supplementary Note 7). Suppose after obtaining the estimate $t_{\mathrm{est}}$ one prepares $m(n)$ copies of state $e^{-iHt_{\mathrm{est}}}|\phi_{\mathrm{coh}}\rangle$. Then, the entire measure-and-prepare process will be described by a TI operation. Furthermore, as we show in Supplementary Note 7, applying this TI operation on the input $\rho^{\otimes n}$, the fidelity of the resulting state with the desired state $|\phi_{\mathrm{coh}}\rangle^{\otimes m(n)}$ is

$$\int_0^\tau dt_{\mathrm{est}} p(t_{\mathrm{est}}|t = 0)|\langle\phi_{\mathrm{coh}}|e^{iHt_{\mathrm{est}}}|\phi_{\mathrm{coh}}\rangle|^{2m(n)} \geq 1 - m(n)F_H(\phi_{\mathrm{coh}}) \times \langle\delta t^2\rangle/4, \qquad (14)$$

where $F_H(\phi_{\mathrm{coh}})$ is four times the energy variance of $\phi_{\mathrm{coh}}$, and $\langle\delta t^2\rangle = \int_0^\tau dt_{\mathrm{est}} p(t_{\mathrm{est}}|t)(t - t_{\mathrm{est}})^2$ is the Mean Squared Error (MSE) of the estimator (Note that because of time-translation symmetry, MSE is independent of $t$). Therefore, the ratio of the

yield $r(n) = m(n)/n$ to infidelity $\epsilon_n$, satisfies

$$\frac{r(n)}{\epsilon_n} \geq \frac{4}{F_H(\phi_{\mathrm{coh}}) \times n\langle\delta t^2\rangle}. \qquad (15)$$

For any reasonable estimator the MSE $\langle\delta t^2\rangle$ scales as $1/n$. Therefore, as $n$ goes to infinity, the above lower bound remains positive. In particular, as shown in[30,70], there exists an estimator working based on the classical Maximum Likelihood (ML) estimator, which achieves MSE $\langle\delta t^2\rangle = 1/(nF_H(\rho)) + o(1/n)$, i.e., saturates the Quantum Cramér-Rao bound[27,30,71]. Therefore, using Eq. (15), we find that the ratio $r(n)/\epsilon_n$ for this estimator, satisfies the lower bound in Eq. (13).

It is worth noting that in the high noise regime, where each input copy $\rho$ is close to the maximally mixed state, we have $P_H(\rho)/F_H(\rho) \approx 1$, and therefore the lower and upper bounds in Eq. (13) coincide, up to a constant factor $1/c$. Therefore, in this regime we can achieve close to optimal distillation using a measure-and-prepare strategy. Furthermore, because asymptotically the optimal MSE can be achieved using local adaptive measurements on individual copies[30,70], this distillation process does not require any entangling interactions between the input copies. On the other hand, as we discuss in Methods section, such measure-and-prepare TI operations are, in general, sub-optimal for distillation in the low-noise regime.

**Single-shot Coherence Distillation: Exact formula.** Next, we consider the problem of coherence distillation in the single-shot regime: suppose we are given $n$ copies of a system in a mixed state $\rho$ as the resource, and we want to obtain a single copy of a system in a pure coherent state $\psi$, using only TI operations? What is the maximum achievable fidelity $\max_{\mathcal{E}_{\mathrm{TI}}} \langle\psi|\mathcal{E}_{\mathrm{TI}}(\rho^{\otimes n})|\psi\rangle$, where the maximization is over all TI operations.

Using the approach of[72], we find a simple general formula for the maximum achievable fidelity:

$$\max_{\mathcal{E}_{\mathrm{TI}}} \langle\psi|\mathcal{E}_{\mathrm{TI}}(\rho^{\otimes n})|\psi\rangle = 2^{-H_{\min}(\mathrm{out}|\mathrm{in})_\Omega}, \qquad (16)$$

where $H_{\min}(\mathrm{out}|\mathrm{in})_\Omega$ is the conditional min-entropy[56,73], for the bipartite state $\Omega_{\mathrm{in,out}}$, obtained by dephasing state $(\rho^{\otimes n})_{\mathrm{in}} \otimes |\psi\rangle\langle\psi|_{\mathrm{out}}$ in the eingenbasis of Hamiltonian $H_{\mathrm{in}} \otimes I_{\mathrm{out}} - I_{\mathrm{in}} \otimes H_{\mathrm{out}}$. Here, $I_{\mathrm{in}}$ and $I_{\mathrm{out}}$ are the identity operators, and $H_{\mathrm{in}} = \sum_{i=1}^n H^{(i)}$ and $H_{\mathrm{out}}$ are the input and output Hamiltonians, respectively. See Supplementary Note 9, for the proof and further discussion about this formula.

Although important, Eq. (16) does not clearly show the asymptotic behavior of the maximum achievable fidelity. On the other hand, our results on the purity of coherence and sub-linear coherence distillation yield simple general upper and lower bounds on the maximum achievable fidelity. Note that in Eq. (15), the number of distilled copies $m(n)$ is arbitrary and can be independent of $n$. In fact, as we explain in Supplementary Note 7, for any (fixed) finite $m(n) = m$, Eq. (15) is tight in the regime $n \to \infty$, and for ML estimator, $n \times \epsilon_n$ converges to $mF_H(\phi_{\mathrm{coh}})/4F_H(\rho)$, where $\epsilon_n$ is the infidelity of the output with $m$ copies of $\phi_{\mathrm{coh}}$.

**Example: Single-shot distillation of a two-level system.** The smallest quantum clock is a system with two different energy levels. Without loss of generality we assume the Hamiltonian of this system is $H = \pi\sigma_z/\tau$. Suppose we want to prepare this clock in state $|\phi_{\mathrm{coh}}\rangle = (|0\rangle + |1\rangle)/\sqrt{2}$, but we have access to a noisy version of this state, i.e., $\rho = \lambda|\phi_{\mathrm{coh}}\rangle\langle\phi_{\mathrm{coh}}| + (1 - \lambda)I/2$, with $0 < \lambda < 1$. The goal is to use $n \gg 1$ copies of $\rho$ to obtain a state with higher fidelity with $|\phi_{\mathrm{coh}}\rangle$. What is the lowest achievable infidelity? Using the properties of the purity of coherence and, in

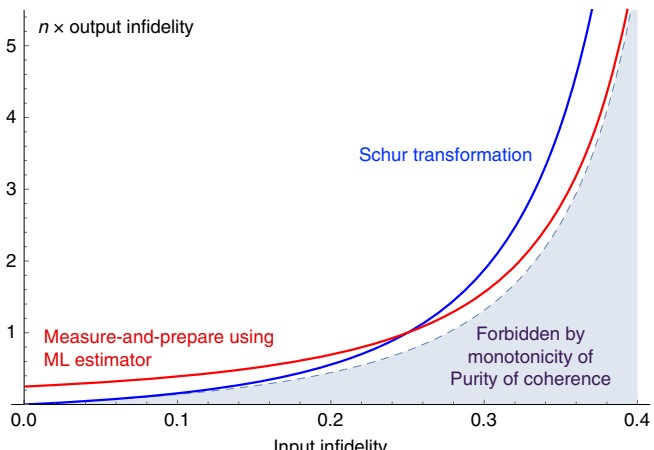

**Fig. 3 Minimum achievable infidelity as a function of the input infidelity.** We are given $n \gg 1$ two-level systems, each with Hamiltonian $\pi\sigma_z/\tau$, in state $\rho = \lambda|\phi_{\text{coh}}\rangle\langle\phi_{\text{coh}}| + (1-\lambda)I/2$, where $0 < \lambda < 1$, i.e., a noisy version of state $|\phi_{\text{coh}}\rangle = (|0\rangle + |1\rangle)/\sqrt{2}$. The goal is to distill a single copy of $|\phi_{\text{coh}}\rangle$ with higher fidelity using TI operations. Horizontal axis is the infidelity of each input state $\rho$ with the desired state $|\phi_{\text{coh}}\rangle$, which is equal to $(1-\lambda)/2$. For any reasonable coherence distillation process, the infidelity at the output is in the form $e(\lambda)/n + o(1/n)$. Vertical axis is the function $e(\lambda)$, i.e., $n$ times the output infidelity, in the limit $n \to \infty$. The dashed curve corresponds to the equation $e(\lambda) = (1-\lambda^2)/4\lambda^2$, dictated by the conservation of purity of coherence (RLD Fisher information), i.e., is found by minimizing the infidelity with the desired state $|\phi_{\text{coh}}\rangle$, under the constraint that the purity of coherence remains conserved. The shaded area below this curve is forbidden by the monotonicity of this quantity. The blue curve is $e(\lambda) = (1-\lambda)/2\lambda^2$, achieved by a distillation process which works based on the Schur transformation[75]. The red curve is $e(\lambda) = 1/4\lambda^2$, achieved by a measure-and-prepare process which uses ML estimator. Note that the lower bound imposed by the purity of coherence is tight in both high-noise ($\lambda \to 0$) and low-noise ($\lambda \to 1$) regimes, but each of these TI operations achieves this lower bound only in one limit.

particular, Eq. (7), in Supplementary Note 10 we show that the infidelity is lower bounded by

$$1 - \max_{\mathcal{E}_{\text{TI}}}\langle\phi_{\text{coh}}|\mathcal{E}_{\text{TI}}(\rho^{\otimes n})|\phi_{\text{coh}}\rangle \geq \frac{1}{n}\frac{1-\lambda^2}{4\lambda^2} + \mathcal{O}\left(\frac{1}{n^2}\right). \quad (17)$$

Therefore, in the limit of large $n$, infidelity times $n$ is lower bounded by $(1-\lambda^2)/4\lambda^2$. In Fig. 3 we compare this lower bound with the infidelity achieved by two different TI processes: (i) an operation related to quantum Schur transformation, studied previously in[74], which has full SU(2) symmetry, and hence is also TI. As we discuss in Supplementary Note 10, the results of[74] implies that using this process we can achieve the infidelity $(1-\lambda)/2n\lambda^2$. (ii) The measure-and-prepare process based on the ML estimator, discussed in the previous section, which achieves the infidelity $n^{-1} \times F_H(\Phi)/4F_H(\rho) = 1/(4n\lambda^2)$.

Remarkably, we find that the bound imposed by the purity of coherence in Eq. (17) is tight in both high-noise ($\lambda \to 0$) and low-noise ($\lambda \to 1$) regimes. This suggests that this bound is achievable for all values of $\lambda$, and, at least in this example, the purity of coherence determines the ultimate limit of coherence distillation in the single-shot regime.

**Discussion.** In recent years there has been a significant progress in understanding the concept of coherence in the context of quantum thermodynamics (See e.g.,[1–4,6,40,75,76]). Nevertheless, some aspects of coherence are still not well-understood. Here, we

highlighted an important feature of quantum coherence which manifests itself, for instance, in the unreachability of pure coherent states from mixed states in both the single-shot and asymptotic regimes, and the fact that (in some precise sense) the coherence content of a single qubit can be arbitrarily large. To quantify this feature of coherence, we introduced a new quantifier of coherence, called the purity of coherence and showed that the monotonicity of this quantity under TI operations gives a tight bound on the coherence distillation in the single-shot regime. The tightness of this bound supports the idea that the purity of coherence is adequately quantifying the unreachability of pure coherent states from mixed states.

In this paper, we focused on the implications of our results in the context of quantum clocks and thermodynamics. Another important area of applications is quantum metrology[32,47,77–79], which will be discussed in future works.

## Methods

**Limited power of TI measure-and-prepare processes for distillation.** In the above example, it is interesting to note that in the high noise regime, the optimal distillation can be achieved using a measure-and-prepare TI process. On the other hand, in the opposite limit, where the input state $\rho$ is almost pure, measure-and-prepare TI processes are not optimal for coherence distillation. In fact, as it can be seen in Fig. 3, even if the input is $n$ copies of a pure coherent state $\phi_{\text{coh}}$, the output of a measure-and-prepare distillation process can not be a pure coherent state for any finite $n$.

To understand this fact better, in the following we derive a strong constraint on the power of measure-and-prepare TI processes for manipulation of coherence. This constraint is a corollary of the following result: For any state $\rho$ and any Measure-and-Prepare TI process $\mathcal{E}_{\text{MP–TI}}$, it holds that

$$P_{H_{\text{out}}}(\mathcal{E}_{\text{MP–TI}}(\rho)) \leq F_{H_{\text{in}}}(\rho) \leq P_{H_{\text{in}}}(\rho), \quad (18)$$

i.e., the purity of coherence of the output is upper bounded by the input QFI, where $H_{\text{in}}$ and $H_{\text{out}}$ are, respectively, the input and output Hamiltonians (See below for further discussion). This means that for input $\rho^{\otimes n}$, the purity of coherence of the output of a measure-and-prepare TI process is upper bounded by $n \times F_{H_{\text{in}}}(\rho)$. On the other hand, for a general TI process the purity of coherence of the output can be as large as $n \times P_{H_{\text{in}}}(\rho)$, which is much larger than $n \times F_{H_{\text{in}}}(\rho)$, if $\rho$ is close to a coherent pure state (For instance, in the above example, Schur transformation reaches this bound in the low noise regime).

Combining this result with the lower bound on the purity of coherence in Eq. (11), we find that if one applies a measure-and-prepare TI process to $n$ copies of $\rho$ to obtain $m(n)$ copies of a pure coherent state $\phi_{\text{coh}}$ with error $\epsilon_n$, then for sufficiently large $m(n)$ and small error $\epsilon_n$, the yield $r(n) = m(n)/n$ and error $\epsilon_n$ satisfy $r(n)/\epsilon_n \leq 1/c \times F_H(\rho)/F_H(\phi_{\text{coh}})$. Therefore, if QFI of state $\rho$ is finite, which is always the case for systems with bounded Hamiltonians, then using measure-and-prepare TI processes it is not possible to achieve a finite yield $r(n) > 0$ with a vanishing error $\epsilon_n \to 0$, even if $\rho$ is a pure coherent state, i.e., has an unbounded purity of coherence.

In Supplementary Note 8 we present the proof of inequality $P_{H_{\text{out}}}(\mathcal{E}_{\text{MP–TI}}(\rho)) \leq F_{H_{\text{in}}}(\rho)$ in Eq. (18). We also note that this inequality follows from the previous result of[80]. The main idea is the following: By definition any measure-and-prepare process can be realized by a measurement on the input followed by a state preparation at the output, which solely depends on the classical outcome of the measurement. For input states $\{e^{-iH_{\text{in}}t}\rho e^{iH_{\text{in}}t}\}_t$, consider the distribution of outcomes of this measurement, as a function of parameter $t$. Then, the (classical) Fisher information corresponding to parameter $t$ is upper bounded by QFI of the input state, i.e., $F_{H_{\text{in}}}(\rho)$. As we show in Supplementary Note 8, this classical Fisher information, itself, is an upper bound on $P_{H_{\text{out}}}(\mathcal{E}_{\text{MP–TI}}(\rho))$, the purity of coherence of the output (This also has been shown previously in[80]). Roughly speaking, this is true because at the classical level, the distinction between Fisher information and the purity of coherence vanishes (This is related to Čencov's theorem[63] which asserts that, up to a normalization, Fisher information is the unique monotone metric on the space of classical probability distributions).

**Violation of monotonicity of Petz-Rényi relative entropy in the light of coherence distillation.** Our results on coherence distillation, and in particular Eq. (13) and Eq. (17), provide a novel operational understanding of the violation of monotonicity of Petz-Rényi relative entropy under data-processing, for $\alpha > 2$. Recall that for $\alpha > 1$, Petz-Rényi relative entropy is defined as $D_\alpha(\rho \parallel \sigma) = \frac{1}{\alpha-1}\log\text{Tr}(\rho^\alpha\sigma^{1-\alpha})$, if $\text{supp}(\rho) \subseteq \text{supp}(\sigma)$ and $D_\alpha(\rho \parallel \sigma) = \infty$, otherwise[55,56]. For $\alpha \in (1, 2]$, and any completely positivity trace-preserving map $\mathcal{E}$, $D_\alpha(\mathcal{E}(\rho) \parallel \mathcal{E}(\sigma)) \leq \mathcal{D}_\alpha(\rho \parallel \sigma)$, whereas this bound is violated for $\alpha > 2$[55,56]. As we mentioned before, the purity of coherence can be derived from the second derivative of the Petz-Rényi relative entropy for $\alpha = 2$, and its monotonicity under

TI operations follows from the monotonicity of this relative entropy (See Supplementary Note 2). Considering the second derivative of Petz-Rényi relative entropy for other values of $\alpha \in (1, \infty)$, we can generalize the purity of coherence, and obtain the family of functions defied by the formula $P_{H,\alpha}(\rho) \equiv \mathrm{Tr}(\rho^{\alpha} H \rho^{1-\alpha} H) - \mathrm{Tr}(\rho H^2)$, if the projector to the support of $\rho$ commutes with $H$, and $P_{H,\alpha}(\rho) = \infty$ otherwise. Similar to the purity of coherence, all these functions are (i) additive, (ii) non-zero iff state is coherent, and (iii) bounded if the projector to the support of $\rho$ commutes with $H$. Furthermore, for any state $\rho$ whose infidelity with a pure coherent state is $\epsilon$, $P_{H,\alpha}(\rho)$ scales (at least) as $\epsilon^{1-\alpha}$. It follows that, if instead of the purity of coherence we use other monotone functions in this family, we obtain other lower bounds on the achievable infidelity. In particular, such a bound would imply that if the purity of coherence of a mixed state $\rho$ is finite, then to distill a single copy of a pure coherent state $\phi_{\mathrm{coh}}$ with error $\epsilon$, the required number of copies of $\rho$ is, at least, of order $\epsilon^{1-\alpha}$, i.e., $n \in \Omega(\epsilon^{1-\alpha})$. For $\alpha > 2$ this bound is asymptotically stronger than the bound imposed by purity of coherence, which is linear in $\epsilon^{-1}$.

However, as we have seen in the proof of Eq. (13) and also in Fig. 3, there exists a TI process based on the ML estimator which achieves errors of order $\epsilon$, by consuming only order $\epsilon^{-1}$ copies of $\rho$. Therefore, if Petz-Rényi relative entropy was monotone for $\alpha > 2$, we had a lower bound on the number of required copies, which was violated by this coherence distillation process. This provides an operational explanation that why the Petz-Rényi relative entropy cannot be monotone under data-processing for $\alpha > 2$: $\alpha = 2$ is the largest value for which the monotonicity of Petz-Rényi relative entropy is not violated by coherence distillation processes.

**Proofs**. All the results in the paper are rigorously proven in the Supplementary Notes 1-10.

## Data availability
Data sharing not applicable to this article as no datasets were generated or analyzed during the current study.

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

## Acknowledgements

I am grateful to Gilad Gour and David Jennings for reading the manuscript carefully, and providing many useful comments and suggestions. Also, I would like to thank Anna Jenčová, Milán Mosonyi, and Keiji Matsumoto for helpful discussions on Fisher Information.

## Author contributions

I.M. was the sole contributor to all the aspects of this work.

## Competing interest

The author declares no competing interest.
