## [Peer Review File · Nature Communications]

Reviewers' comments:

Reviewer #1 (Remarks to the Author):

In the manuscript "Coherence distillation machines are impossible in quantum thermodynamics" the author studies so called "coherence distillation machines", which are hypothetical devices extracting quantum coherence from noisy quantum systems. The main results of the paper are:

- 1) Coherence distillation machines do not exist, in the sense that there is no device which creates pure coherent states from a noisy system at a finite rate.
- 2) Coherence amplification is still possible in some settings: particular noisy quantum states can be "purified" into states having coherence, however with a sublinear rate.

The questions raised in the paper are timely: in fact most of the research results on coherence distillation has focused on the so-called "speakeable coherence", where permutations of energy eigenstates are considered as free transformations. The author of submitted manuscript significantly contributes to this discussion by presenting results on distillation of "unspeakeable coherence": here the free transformations are translationally invariant. Apart from its elegant definition, this concept has proven very useful also for quantum thermodynamics.

Indeed, the findings 1) and 2) mentioned above are remarkable and (in this combination) counterintuitive, and lead to several follow up questions:

- Around Eq. (8) the author shows in an explicit example, where a mixed single-qubit state can be asymptotically converted into one copy of a maximally coherent state. Can every state which has coherence be converted into a maximally coherent state in this way, or is there any phenomenon similar to bound coherence in this setting?

- Moreover, can the author give a functional dependence on the number of extractable maximally coherent states (e.g. whether the number is logarithmic in the number of initial states ρ)? I agree that an exact solution of this problem for all states might be intractable, but one would hope to solve it exactly at least for some families of states.

Further comments:

- In Eq. (1) and Fig. 2, the author distinguishes between different Hamiltonians for the input and the output system. From the first glance it is not clear why this is required, as the translationally invariant operations are usually defined with the same Hamiltonian. As I understand, the distinction is required here because the input and the output systems have different dimensions. Could the author clarify this point?

- When the author defines an "ideal coherence distillation machine" on page 3, why does the author assume that ϕ_{coh} is an arbitrary state, except for the energy eigenstates? As I understand, the author aims to show that the distillation is not working for any state which has coherence. However, at this point it would be more natural to set ϕ_{coh} to some specific states, for example those with maximal coherence.

- Also on page 3, the author writes: "...we can allow a small error in the trace distance (l_1 -norm)..."

This is not correct, as the trace norm is not the same as l_1 -norm.

- The theorem on page 3 seems to come out of the blue, a better connection to the previous paragraph would be desirable.

In summary, the submitted article is timely and interesting, and also introduces novel techniques for studying extraction of quantum coherence (purity of coherence). However, I would postpone my final judgement on the manuscript, hoping that the author can address all questions above.

Coherence distillation machines are impossible in quantum thermodynamics – Referee report Nature Communications

I. GENERAL COMMENTS

The present paper deals with the problem of coherence distillation under time-translationally invariant (TI) operations. The setting is as follows: one consider a large number of microscopic “clocks”, all initially prepared in the same quantum state and evolving independently according to their own Hamiltonian. These are our reference systems, i.e. they define the way we measure time. To get rid of the noise and obtain a more precise device we may want to carry out some sort of operation on all these systems jointly, with the goal of outputting some pure states, which – being noiseless – can track time more precisely. Since the input systems identify our very definition of time, the only operations that we are allowed to use are those that can be implemented without access to any clock; these are exactly the aforementioned TI operations.

The main result of the paper is that such a distillation process is impossible for most initial mixed states if one requires the number of output states to grow linearly with the number of inputs. This means that we either accept a sublinear scaling of the number of pure clocks produced, or we give up on trying to reduce the distillation error to zero. I find the setting quite natural and of potential relevance for near-term applications. Also, the result is undoubtedly interesting, for: (i) it affects an operationally motivated resource theory in a decisive way; (ii) it shows a peculiar feature of TI operations, encouraging the community to work towards extending that framework; (iii) it sheds light on other aspects of the recently investigated phenomenon of bound coherence, which seems to have attracted considerable attention; (iv) more at the level of mathematical curiosity, it gives us an example of a resource theory where distillation with nonzero rate is often impossible, yet single-copy distillation with vanishing error is allowed. Moreover, the techniques used throughout the paper are innovative under certain aspects. I checked all the mathematical proofs and found no mistakes other than superficial typos.

However, there are also downsides. I am a bit surprised to see that not all of the results in the preprint arXiv:1805.01989 have been reported here. This is particularly disappointing as the other main result in that preprint nicely complements the one given here; namely, it shows that the coherence cost of any state is a multiple of its Fisher information, thus giving an operational interpretation to this latter quantity. Since distillation and formation are complementary processes in any resource theory, I find a bit unnatural that these two results be separated; ideally, they should be discussed together. If the author is willing to include them here, then I am ready to recommend publication of what would become a very strong submission. Otherwise he should give convincing reasons why this is not appropriate.

In conclusion, I believe that the paper presents a nontrivial results that is potentially of strong interest. However, to complete the picture of “unspeakable” coherence transformations and thus meet the high standards of Nature Communications, the other results on TI coherence cost should perhaps be included. Below there are some more comments and a list of typos that I found while reading the manuscript.

- Just out of curiosity, can one give a closed formula to compute the TI distillable coherence for those states that happen to be distillable?
- It is unclear to me how the results here compare to those on the SIO bound coherence (in the “speakable” setting), see [arXiv:1808.01885, arXiv:1809.06880, and arXiv:1902.02427]. A more detailed discussion of this point should be given, the single sentence in the current version is not really of much help.

II. MAIN TEXT

1. First column p.1: avoid repetition of word “question”, for instance by substituting “the answer to both questions is negative.” → “the answer to both of them is negative”
2. Eliminate double space after “i.e.”, “e.g.” and so on.
3. After Eq. (3b): the support is usually denoted with “supp” rather than “sup”, to avoid confusion with the supremum.
4. “Renyi” → “Rényi”, because accents are important in Hungarian.
5. Last line second column p.3: “a convex function of ρ ”.
6. Maybe at some point in the discussion of the properties of the purity of coherence in the main text it should be said that it is always non-negative, which helps the reader make sense of property (i), i.e. faithfulness. Clearly, positivity follows easily from the connection with the Petz-Rényi relative entropy of order 2 discussed in the SM, but it is not obvious from the definition in Eq. (3a)–(3b).
7. “Milan Mosonyi” → “Milán Mosonyi”; again, in Hungarian there are accents.

III. SUPPLEMENTARY MATERIAL

8. Please check the spelling of names throughout: “Steinspring” → “Stinespring”.
9. Remark 2: “check that a quantum operation is completely-incoherence preserving or not” → “check whether a quantum operation is completely-incoherence preserving or not”.
10. Remark 2: avoid repetition of “check this condition”.
11. p.2: “According this lemma” → “According to this lemma”.
12. Right after Eq. (6), the set of time-evolved Kraus operators lacks dependence on t . Also, the last inline equation of the paragraph should have μ' inside the summation.
13. After Eq. (14): what is “the second third line” ???
14. Before Eq. (28): “can written as” → “can be written as”.
15. Define function f_H in Eq. (46).
16. After Eq. (65): “completely-positive” → “completely positive”.
17. The proof of Lemma 4 seems odd to me. It is very clear that the support of $HPi_\rho H$ is the same as that of $H\rho H$, simply because the kernels of the two operators are the same. Then the first two-and-a-half lines can be skipped entirely.
18. Last line p.10: why is “Completely Positive Trace Preserving” now uppercase? Please be consistent with your conventions.
19. After Eq. (72): “satisfies” → “satisfy”.

20. The calculation in Eq. (75)–(78) replicate those in Eq. (60)–(62). Please eliminate this redundancy.
21. The inequality between purity of coherence and quantum Fisher information follows straightforwardly from the arithmetic-geometric mean inequality applied to p_j and p_k . This allows to get rid of the calculations in Eq. (80)–(84).
22. Not once has the variance of the Hamiltonian been defined. Thus Eq. (90) is ambiguous.
23. Eq. (111) should not be preceded by “This implies”, as it is always valid.
24. The proof of Theorem 8 is wonderful and its exposition quite intuitive. I wonder whether it is even too intuitive, and its proof should be made more formal.
25. One should define $|\bar{\psi}\rangle$ straight after Eq. (167), so that readability is not compromised.

Reviewer #3 (Remarks to the Author):

Referee report for the manuscript entitled “Coherence distillation machines are impossible in quantum thermodynamics” (identifier NCOMMS-18-28790726-T) by Iman Marvian, submitted for consideration as an article in Nature Communications.

The manuscript discusses the possibility of distilling pure states featuring non-vanishing off-diagonal elements w.r.t. the energy eigenbasis (pure coherent states) from mixed states with non-vanishing off-diagonal elements with respect to the energy eigenbasis (mixed coherent states) at a nonzero rate using time-translation invariant (TI) operations. The main result of the paper is a no-go result (as stated in the manuscript): “... starting from asymptotically many copies of a generic mixed state, using a thermal machine we cannot distill pure coherence at a nonzero rate, even if we spend an unlimited amount of work.” Besides this main result, the manuscript also discusses the equivalence between TI operations, completely incoherence preserving operations, and operations that are implementable with thermal operations supplemented by an arbitrary amount of work. The manuscript relates various quantifiers of the coherence in question, in particular, to the quantum Fisher information, and the relation to other approaches to resource theories of coherence is briefly mentioned.

The manuscript is overall well-written in clear scientific English (apart from some very minor typos and grammatical mistakes like missing articles here and there) and is well structured. A lot of the mathematical details are stowed away in a supplemental document. One may argue that this has the advantage of the manuscript being easily readable, but at least in my personal opinion, I would prefer if these details were presented in the same file, possibly as an appendix, rather than in an entirely different document (that is not being copy-edited). In any case, the results appear to be technically sound and the paper supports its conclusions with evidence (again, in the supplemental material).

Despite there being a number of other articles in the field about the various resource theories of coherence, and also about coherence distillation, the results obtained in this particular paradigm TI/thermal operations appears to be novel and I suppose these results will be interesting to researchers interested in coherence as a resource, and influence the thinking of these researchers.

That being said, I have some critical remarks I would like to make:

(1) The premise of the paper appears to be the distillation of “pure clocks” from “mixed clocks” and this connection is established via the observation that states useful for keeping time must be time-dependent, and must hence feature off-diagonals with respect to the system Hamiltonian. However, apart from this remark, the relevance of this distillation procedure to the performance of clocks is not explained at all. Why is it better to have a pure clock state than a mixed clock state, quantitatively? I think, to the reader with some background on quantum clocks and metrology, the section on the relation to the quantum Fisher information can give some clues, but at least in the introduction the particular relevance of coherence distillation to obtaining “better” clocks is somewhat obscure. I think that this aspect should be revised.

(2) While I believe that the article will be of interest to researchers working on coherence as a resource, to me, the concept of a resource theory of coherence seems rather strange. Mathematically, resource theories are nice framework of phrasing problems. For some of them, e.g., resource theory of entanglement, it makes a lot of sense, practically speaking, to formulate entanglement as a resource, distributing entangled states across distances is difficult and costly, and so it is of practical relevance to understand what can be done by means of local operations and classical communication (LOCC). In other words, there is a practical notion for entanglement being a resource, which is the premise for assuming that LOCC operations are free. For coherence as a resource, this does not seem to me to be so clear. In typical laboratories, coherence does not seem to be so difficult to create. So, while I understand the fundamental interest from a mathematical point of view, the practical relevance of the resource theory of coherence seems rather doubtful. For the present manuscript, I therefore think that the investigating of these hypothetical coherence distillation machines is somewhat ill motivated. I do however believe that the result is interesting from the point of view of quantum thermodynamics, in the sense that it shows a fundamental limitation of thermal operations. This remark is mostly expressing my own

opinion on the general topic: As I mentioned, I believe there is a sufficiently large community of researchers working on resource theories of coherence (whether I think it is a good idea or not) to which this work will be of relevance, and one can hence argue the paper satisfies the acceptance criteria of Nature Communications.

Some quick comments:

- In the introduction it says: "Suppose we have multiple clocks, all synchronized with the same reference clock," and later it is stated that operations are to be "without having access to a synchronized clock". It is not clear why a reference clock is first available, and then not available. I think this should be clarified.
- In the introduction it says: "In other words, the coherence distillation machine, depicted in Fig.1, is impossible, which is reminiscent of the impossibility of perpetual motion machines or cloning machines." In what way is it reminiscent other than these being things that are impossible? I think this should be more clearly explained, what are the parallels between these situations other than "things that are not possible"?
- It is further stated that the result "reveals important aspects of coherence in quantum thermodynamics." Which aspects? Without a specific explanation or example, this part of the sentence is rather meaningless.

Some of the typos I spotted:

- Section on Quantum Clocks: "By definition, state of a clock" -> "By definition, the state of a clock"
- Same section: "To study manipulation of clocks" -> "To study the manipulation of clocks"
- "time-Translationally Invariant (TI) operations": I think it should be "time-translation invariant", I don't think the use as an adverb is correct here, the invariance is not time-translational, the operations are invariant under time-translations.
- "these are transformation which satisfy" -> "these are transformations which satisfy"
- "we may need to turn on an interaction and let them to exchange energy." -> "we may need to turn on an interaction and let them exchange energy."
- Section purity of coherence: "In the recent years," -> "In recent years,"

Response to Reviewers

Summary of the main changes:

- 1) To address two questions raised by reviewer #1, about the possibility and limits of sub-linear distillation, I have added a new section to the paper, titled: “*Sub-linear Coherence Distillation: Trade-off between the maximum achievable yield and fidelity*”.
- 2) Motivated by a discussion by reviewer #3, about operational relevance of TI operations and the corresponding resource theory, I have revised the introduction, to explain how the set of TI operations arises naturally in the context of quantum clocks.
- 3) Motivated by recommendation of reviewer #2, I have presented the proof of the main theorem in the SM more formally, and have also updated the corresponding section in the main paper to make it more formal.
- 4) In the last section, on single-copy distillation, I have added a figure which compares the achievable infidelity using Schur transformation with the lower bound imposed by the purity of coherence (In the previous version, the formulas were presented in the text, without the graph). This graph also shows the achievable infidelity, using a measure-and-prepare strategy.

I have also implemented a few cosmetic changes at various points in the paper where I thought that an idea was not expressed as clearly as it could have been.

Response to Reviewer #1

I would like to thank the referee for the positive review and the great questions.

1)

- Around Eq. (8) the author shows in an explicit example, where a mixed single-qubit state can be asymptotically converted into one copy of a maximally coherent state. Can every state which has coherence be converted into a maximally coherent state in this way, or is there any phenomenon similar to bound coherence in this setting?

- Moreover, can the author give a functional dependence on the number of extractable maximally coherent states (e.g. whether the number is logarithmic in the number of initial states ρ)? I agree that an exact solution of this problem for all states might be intractable, but one would hope to solve it exactly at least for some families of states.

I have fully addressed both questions in a new section titled “Sub-linear Coherence Distillation: Trade-off between the maximum achievable yield and fidelity”.

In summary, the answer is *any* state which contains coherence can be used to distill maximally coherent state with arbitrary small error, provided that one can consume sufficiently many copies. Furthermore, there is a tradeoff between the number of distilled copies and the achievable fidelity. In particular, if the ratio of the distilled copies of maximally coherent states to the input copies vanishes, then the state conversion can be implemented using a TI operation, with a vanishing error.

2)

- In Eq. (1) and Fig. 2, the author distinguishes between different Hamiltonians for the input and the output system. From the first glance it is not clear why this is required, as the translationally invariant operations are usually defined with the same Hamiltonian. As I understand, the distinction is required here because the input and the output systems have different dimensions. Could the author clarify this point?

This generality is crucial for coherence distillation, because the number of input and output copies are in general different. Therefore, the input and output have different Hilbert spaces and different Hamiltonians.

I have revised the introduction to clarify this and also address a concern by referee 3.

3)

- When the author defines an "ideal coherence distillation machine" on page 3, why does the author assume that ϕ_{coh} is an arbitrary state, except for the energy eigenstates? As I understand, the author aims to show that the distillation is not working for any state which has coherence. However, at this point it would be more natural to set ϕ_{coh} to some specific states, for example those with maximal coherence.

As the referee has mentioned, here I am trying to emphasize that the distillation process does not work, regardless of the choice of coherent pure state. Also, I have been trying to write the paper such that a general reader who is not necessarily familiar with resource theories and the notion of "maximal coherence" can understand the main surprising result. So, I have been trying to avoid using concepts which are not crucial for understanding the main result.

To address the referee's concern, I have added this sentence: "For instance, one can choose a two-level system with Hamiltonian $\pi \sigma_z / \tau$, and state $|\phi_{\text{coh}}\rangle = (|0\rangle + |1\rangle) / \sqrt{2}$, where τ is the period."

4)

- Also on page 3, the author writes: "...we can allow a small error in the trace distance (l1-norm)..." This is not correct, as the trace norm is not the same as l1-norm.

I switched this to infidelity. Since in the example on single-copy distillation, I talk about infidelity, for consistency I decided to use infidelity as the quantifier of error in the paper. Given that the trace distance vanishes iff infidelity vanishes, this does not affect the definition.

5)

- The theorem on page 3 seems to come out of the blue, a better connection to the previous paragraph would be desirable.

I added the following sentence before theorem: “We prove the following fundamental no-go theorem on coherence distillation”

Response to Reviewer #2

I would like to thank the referee the positive review. Especially, I am grateful for checking all the proofs in the Supplementary Material. I really appreciate the effort and time.

1)

I am a bit surprised to see that not all of the results in the preprint arXiv:1805.01989 have been reported here. This is particularly disappointing as the other main result in that preprint nicely complements the one given here; namely, it shows that the coherence cost of any state is a multiple of its Fisher information, thus giving an operational interpretation to this latter quantity. Since distillation and formation are complementary processes in any resource theory, I find a bit unnatural that these two results be separated; ideally, they should be discussed together. If the author is willing to include them here, then I am ready to recommend publication of what would become a very strong submission. Otherwise he should give convincing reasons why this is not appropriate.

I understand the referee's point and truly appreciate her/his concern to improve this paper. Here, I explain the main reasons for excluding those discussions from this paper:

Normally, in the resource theory papers, the iid and single-shot regimes are discussed in separate papers. But, in this case, because of the peculiar features of this resource theory, I think it makes sense to devote one paper to distillation and discuss both iid and single-shot regimes side-by-side, and compare them with each other. In other words, rather than dividing the results in terms of iid versus single-shot line, in my opinion, in this case it is better to divide them in terms of distillation vs formation (Please note that, in the revised version, to address some questions raised by reviewe#1, I have added further discussions about sublinear distillation).

Also, please note that the result on formation and coherence cost is related to several other results, which should all be discussed together (e.g. a converse bound for QFI which is useful and relevant for other applications, a new proof of Toth-Petz conjecture, the details of pure state to pure state transformation,...). Given the importance of Quantum Fisher Information in Metrology, these results could be useful for a larger community. Therefore, I'd rather not to put all these materials in an extremely long Supplementary Material (Currently SM has over 40 pages, adding these materials makes it over 65 pages).

Finally, another advantage of the current format is that it allows me to present the main surprising result, which is understandable for anyone with an elementary background in Quantum Information, immediately after defining the framework. This makes the results accessible to a larger community which Nature Communications is aiming at (The resource theory papers usually start with the study of pure state to pure state transformations, and definition of standard unit of resource. However, to understand that the distillation rate is zero, the reader does not need to go through this long path).

In summary, given the above reasons, I prefer to keep the format of the paper unchanged and leave the results on coherence cost out of this paper.

2)

Just out of curiosity, can one give a closed formula to compute the TI distillable coherence for those states that happen to be distillable?

This is a great question. I do not know the answer in general. In the resubmitted version, I have considered a special example in SM (Section E), and have shown an achievable rate which is determined by function $\text{Tr}(H \rho H \Pi_{\perp})$, where Π_{\perp} is the projector to the kernel of ρ . This function can be obtained from Petz-Rényi Relative entropy and is additive and monotone, and non-zero iff purity of coherence is infinite. In other words, it has all the right properties that distillable coherence needs to have. However, currently I cannot generalize this example.

3)

It is unclear to me how the results here compare to those on the SIO bound coherence (in the “speakable” setting), see [arXiv:1808.01885, arXiv:1809.06880, and arXiv:1902.02427]. A more detailed discussion of this point should be given, the single sentence in the current version is not really of much help.

I added the following comment about distillation under SIO to the paper:

“In particular, it is interesting to note that, under the class of Strictly Incoherent Operations \cite{winter2016operational, yadin2016quantum}, the distillable coherence for full-rank states is zero \cite{lami2018generic, regula2018one, zhao2018one}. However, in contrast to the case of TI operations, in this theory there are bound states, which cannot be converted to a single copy of a pure coherent state with a vanishing error, even if one is given an arbitrary many copies of the bound state \cite{zhao2018one}”.

At this point I cannot make any further comment about the possible relations between the two resource theories (Since TIO and SIO behave differently under composition of systems, it is not clear if there is any relation between the two theory. However, I expect that some techniques that are used in this paper can also be used in the case of SIO, with slight modifications).

4)

First column p.1: avoid repetition of word “question”, for instance by substituting “the answer to both questions is negative.” → “the answer to both of them is negative”

I edited this. Thanks for the suggestion.

5)

Eliminate double space after “i.e.”, “e.g.” and so on.

I checked this.

6)

After Eq. (3b): the support is usually denoted with “supp” rather than “sup”, to avoid confusion with the supremum.

I fixed this.

7)

“Renyi” → “R[˘]enyi”, because accents are important in Hungarian.

I corrected this. Thank you for bringing this point to my attention.

8)

Last line second column p.3: “a convex function of ρ ”.

I fixed this.

9)

Maybe at some point in the discussion of the properties of the purity of coherence in the main text it should be said that it is always non-negative, which helps the reader make sense of property (i), i.e. faithfulness. Clearly, positivity follows easily from the connection with the Petz-R[˘]enyi relative entropy of order 2 discussed in the SM, but it is not obvious from the definition in Eq. (3a)–(3b).

I added a couple of sentences immediately after the definition of the purity of coherence, in which I mention the fact that this quantity is non-negative. Also, I explain that this function is a special case of a family of generalized Fisher information, previously introduced by Petz.

10)

“Milan Mosonyi” → “Mil’an Mosonyi”; again, in Hungarian there are accents.

I corrected this.

11)

Please check the spelling of names throughout: “Steinspring” → “Stinespring”.

I corrected this.

12)

Remark 2: “check that a quantum operation is completely-incoherence preserving or not” → “check whether a quantum operation is completely-incoherence preserving or not”.

I fixed this. Thanks.

13)

Remark 2: avoid repetition of “check this condition”.

I edited this.

14)

p.2: “According this lemma” → “According to this lemma”.

I fixed this.

15)

Right after Eq. (6), the set of time-evolved Kraus operators lacks dependence on t . Also, the last inline equation of the paragraph should have μ' inside the summation.

I fixed this. Thank you very much for catching all these typos!

16)

After Eq. (14): what is “the second third line” ???

I corrected this!

17)

Before Eq. (28): “can written as” \rightarrow “can be written as”.

I corrected this.

18)

Define function f_H in Eq. (46).

I added the definition.

19)

After Eq. (65): “completely-positive” \rightarrow “completely positive”.

I fixed this.

20)

The proof of Lemma 4 seems odd to me. It is very clear that the support of $HP_i\rho H$ is the same as that of $H\rho H$, simply because the kernels of the two operators are the same. Then the first two-and-a-half lines can be skipped entirely.

I agree with the referee. I edited this. Thanks for the comment.

21)

Last line p.10: why is “Completely Positive Trace Preserving” now uppercase? Please be consistent with your conventions.

I corrected this.

22)

After Eq. (72): “satisfies” → “satisfy”.

I corrected this.

21)

The calculation in Eq. (75)–(78) replicate those in Eq. (60)–(62). Please eliminate this redundancy.

I corrected this. Thanks for noticing this.

22)

The inequality between purity of coherence and quantum Fisher information follows straight-forwardly from the arithmetic-geometric mean inequality applied to p_j and p_k . This allows to get rid of the calculations in Eq. (80)–(84).

Thanks a lot! I applied it. Also, I mentioned that Quantum Fisher information and the purity of coherence are equal iff the state is incoherent, in which case both quantities are zero.

23)

Not once has the variance of the Hamiltonian been defined. Thus Eq. (90) is ambiguous.

I added the definition to the main paper as well as the supplementary material.

24)

Eq. (111) should not be preceded by "This implies", as it is always valid.

I corrected this.

25)

The proof of Theorem 8 is wonderful and its exposition quite intuitive. I wonder whether it is even too intuitive, and its proof should be made more formal.

I appreciate the positive comment. As referee suggested, I made the proof more formal, and updated the related sections in the paper, and the Supplementary Material (SM). Formalizing the previous argument, in the SM I prove a lower bound on the purity of coherence of states which are close to an idd pure coherent state, i.e. m copies of a pure state which contains coherence, in the limit of large m . This lower bound is now presented in Eq.8b of the resubmitted paper. The proof of the main theorem then follows from this bound together with monotonicity and additivity of the purity of coherence, which is discussed in the main paper.

26)

One should define $|\psi\rangle$ straight after Eq. (167), so that readability is not compromised.

I added the definition after Eq. (167) (Please note in the resubmitted version this is Eq.H4).

Response to Reviewer #3

I thank the referee for the positive report and for good questions and comments, which improved the paper.

1)

While I believe that the article will be of interest to researchers working on coherence as a resource, to me, the concept of a resource theory of coherence seems rather strange. Mathematically, resource theories are nice framework of phrasing problems. For some of them, e.g., resource theory of entanglement, it makes a lot of sense, practically speaking, to formulate entanglement as a resource, distributing entangled states across distances is difficult and costly, and so it is of practical relevance to understand what can be done by means of local operations and classical communication (LOCC). In other words, there is a practical notion for entanglement being a resource, which is the premise for assuming that LOCC operations are free. For coherence as a resource, this does not seem to me to be so clear. In typical laboratories, coherence does not seem to be so difficult to create.

So, while I understand the fundamental interest from a mathematical point of view, the practical relevance of the resource theory of coherence seems rather doubtful. For the present manuscript, I therefore think that the investigating of these hypothetical coherence distillation machines is somewhat ill motivated.

I do however believe that the result is interesting from the point of view of quantum thermodynamics, in the sense that it shows a fundamental limitation of thermal operations.

This remark is mostly expressing my own opinion on the general topic: As I mentioned, I believe there is a sufficiently large community of researchers working on resource theories of coherence (whether I think it is a good idea or not) to which this work will be of relevance, and one can hence argue the paper satisfies the acceptance criteria of Nature Communications.

I would like to thank the referee for sharing her/his point of view on the practical relevance of the resource theories of coherence. I am sympathetic with this point of

view (In fact, I have discussed this exact point in PRA 94, 052324. Please see the section on “*Criticism of resource-theoretic approaches to speakable coherence*”).

However, in the case of TI operations, I believe there are strong operational motivations to study the resource theory. I have revised the introduction to explain this point better. In particular, not only TI operations are relevant in the context of quantum thermodynamics, they also automatically arise in the context of quantum clocks. It follows that the results on distillation and formation in this resource theory, have real physical implications in this context (and this explains why fundamental concepts such as Quantum Fisher Information, find operational interpretation in this resource theory). In the following, I describe the idea briefly. Please see the revised introduction for further details.

Consider the problem of manipulation of clocks, using *arbitrary* Completely Positive Trace-Preserving (CPTP) maps. Suppose, in addition to the given quantum clock, which is the input of the map, we do not have any additional information about the reference clock. This means that to manipulate the clock we have to use a fixed CPTP map, independent of the time parameter. Then, using a simple twirling argument, one can *prove* that the only possible state conversions are those which can be implemented using TI operations. In other words, restriction to the set of TI operations is a consequence of a practical limitation, namely, lack of (additional) synchronized clocks (other than the given quantum clock which is the input of the CPTP map). This can be compared with the restriction to LOCC in entanglement theory, which is a consequence of lack of quantum channels between two distant parties.

I agree with the referee that coherence is not so difficult to create in laboratory. But, in my opinion, resource theories, including thermodynamics and entanglement theory, are about distinguishing between easier and harder operations under certain physical assumptions about available free resources. In general, creating states which contain coherence relative to energy eigenbasis is *harder* than creating incoherent states: In order to create coherent states one needs to interact with a stable reference clock (coherence reservoir), e.g. a laser beam, and any instability in this clock will add noise to state preparation. On the other hand, to prepare incoherent states, interaction with a reference clock is not required. Therefore, in my opinion, the notion of coherence as a resource, defined in terms of TI operations, is relevant from a practical point of view.

Please see the revised introduction for further details.

2)

The premise of the paper appears to be the distillation of “pure clocks” from “mixed clocks” and this connection is established via the observation that states

useful for keeping time must be time-dependent, and must hence feature off-diagonals with respect to the system Hamiltonian. However, apart from this remark, the relevance of this distillation procedure to the performance of clocks is not explained at all. Why is it better to have a pure clock state than a mixed clock state, quantitatively? I think, to the reader with some background on quantum clocks and metrology, the section on the relation to the quantum Fisher information can give some clues, but at least in the introduction the particular relevance of coherence distillation to obtaining “better” clocks is somewhat obscure. I think that this aspect should be revised.

I clarified this in the revised introduction:

“..... In particular, the notion of *\emph{resource distillation}*, which can be abstractly defined in any resource theory, has a clear operational interpretation in this theory: it is the process in which one combines many noisy clocks, affected by independent noise processes, to obtain less, but more accurate clocks in pure states, which contain more information about time (Note that to maximize the information content of the clock, we should prepare it in a pure state). “

3)

• In the introduction it says: “Suppose we have multiple clocks, all synchronized with the same reference clock,” and later it is stated that operations are to be “without having access to a synchronized clock”. It is not clear why a reference clock is first available, and then not available. I think this should be clarified.

I have revised the introduction to clarify this point. In short, we assume other than the given quantum clock, we do not have any additional information about the reference clock. In other words, all the information about the unknown time parameter is contained in the state of the given quantum clock, which is the input of the TI operation.

4)

• In the introduction it says: “In other words, the coherence distillation machine, depicted in Fig. 1, is impossible, which is reminiscent of the impossibility of perpetual motion machines or cloning machines.”

In what way is it reminiscent other than these being things that are impossible? I think this should be more clearly explained, what are the parallels between these situations other than “things that are not possible”?

Thanks for the comment. To clarify this, and to focus on the analogy with impossibility of perpetual motion machines, I edited this sentence and removed the word “cloning machines” :

“In other words, the coherence distillation machine, depicted in Fig.1 is impossible, which is reminiscent of the impossibility of perpetual motion machines (we will discuss more about this analogy later).”

Later in the paper, in Sec. II.C, after discussing the analogy between the free and total energy in thermodynamics on one hand, and the purity of coherence and quantum fisher information, on the other hand, I have added the comment:

“In thermodynamics, consideration of the total energy alone is not sufficient to prove the impossibility of perpetual motion machines. Similarly, the impossibility of the hypothetical coherence distillation machine in Fig.1, cannot be shown using the monotonicity of QFI alone. Rather, to prove this one needs to consider a quantity such as the purity of coherence, which distinguishes pure coherence from the total coherence.”

5)

•It is further stated that the result “reveals important aspects of coherence in quantum thermodynamics.” Which aspects? Without a specific explanation or example, this part of the sentence is rather meaningless.

I added the following sentence to clarify this: “In particular, we will see that, in some precise sense, the coherence content of a single two-level system can be infinitely large.”

6)

Some of the typos I spotted:

• Section on Quantum Clocks: “By definition, state of a clock” -> “By definition, the state of a clock”

I fixed this.

- *“time-Translationally Invariant (TI) operations”*: I think it should be *“time-translation invariant”*, I don't think the use as an adverb is correct here, the invariance is not time-translational, the operations are invariant under time-translations.

I fixed this.

- *“these are transformation which satisfy”* -> *“these are transformations which satisfy”*

I fixed this.

- *Section purity of coherence: “In the recent years,”* -> *“In recent years,”*

I fixed this.

REVIEWERS' COMMENTS:

Reviewer #1 (Remarks to the Author):

The author has convincingly addressed all criticism, and I recommend publication in Nature Communications.

Reviewer #2 (Remarks to the Author):

In this revised version of his manuscript, the author has implemented extensive changes to the text, bringing in new ideas and elaborating on previously introduced concepts. In particular, he has provided a valid justification of why the results in the arXiv preprint do not belong in this paper. This was one of the main concerns I expressed in the previous report, and I feel it has been fully addressed. I am also very happy to see that a great deal of effort has been devoted to improving the exposition of the main theorem's proof, which now meets all standards of rigor.

I checked the added material, and found that all questions I had raised have been satisfactorily answered. I particularly liked the example given in Appendix E of a state with nonzero distillable coherence, and also the interpretation of the violations to the data processing inequality for the Petz-Rényi relative entropies in the parameter range $\alpha > 2$. The newly added section on sublinear coherence distillation, and in particular Eq. (10) is also of great conceptual value, and further contributes to the probable great impact of the paper. All in all, I wholeheartedly recommend publication.

Few small comments and suggestions:

Below Eq. (E1): Instead of "the pure state $|\psi\rangle\langle\psi|$ has support outside this subspace" I would simply say "the pure state $|\psi\rangle$ does not belong to this subspace"

The author could perhaps consider citing [70] (update the reference) and also [arXiv:1902.02427] together with [69] for the discovery and investigation of SIO bound coherence, an example of which had been first provided in [69], but whose complete study and characterization as a generic phenomenon has only been completed in [70] and [arXiv:1902.02427].

Ludovico Lami

Reviewer #3 (Remarks to the Author):

The author has replied to my remarks and made some corresponding changes to the manuscript. Before further commenting on these changes and the author's reply, let me begin by commenting on my overall impression. Judging from the reports of the other referees and the corresponding replies by the author, I feel that I can uphold my previous statement that, even if I might have some reservations as to the motivation and practical relevance of this work, it seems that there is a substantial community of researchers working on or generally interested in resource theories of coherence to whom this work will be of relevance, and to which it makes a useful contribution.

That being said, I see that the author made some changes to the manuscript, in particular to the introduction, to try to alleviate my concerns, but it seems to me that some of these changes might have missed the intent of my questions.

To explain more clearly, let me paraphrase from the author's response:

"However, in the case of TI operations, I believe there are strong operational motivations to study the resource theory. I have revised the introduction to explain this point better. In particular, not only TI operations are relevant in the context of quantum thermodynamics, they also automatically arise in the context of quantum clocks. It follows that the results on distillation and formation in this resource theory, have real physical implications in this context (and this explains why fundamental concepts such as Quantum Fisher Information, find operational interpretation in this

resource theory). In the following, I describe the idea briefly. Please see the revised introduction for further details.”

It seems to me that the main motivation for this resource theory is the stipulation that a number of identical copies of a system state have all been synchronized with a reference clock, and that one does not have access to this reference clock when performing operations on the system. Now, of course such a situation might arise in some cases, and the corresponding resource theory is mathematically clearly outlined, but this does not seem to be a fundamental problem. If one had access to the reference clock, even if it is imperfect, the set of possible operations would be very different. Sure, one is required to have access to this clock, but I am not entirely sure what the big practical (or fundamental) challenge there is to that.

“I agree with the referee that coherence is not so difficult to create in laboratory. But, in my opinion, resource theories, including thermodynamics and entanglement theory, are about distinguishing between easier and harder operations under certain physical assumptions about available free resources. In general, creating states which contain coherence relative to energy eigenbasis is harder than creating incoherent states.”

Personally, I feel that there are indeed big differences between these resource theories. In entanglement theory, once we have spatially separated two parties, we really do not (by any currently known practical means whatsoever) have access to joint quantum operations on these systems. Even most LOCC operations might be in general be difficult to realize in practice, but at least LOCC tells us something about what could in principle at best be achieved with the available means. In quantum thermodynamics, restricting to energy preserving operations is motivated from a fundamental point of view. There, it might be unrealistic to allow for arbitrary energy preserving operations on arbitrarily large and arbitrarily complex baths (originally in thermal states), but again, in principle thermal operations capture what could at best be done in principle. For the resource theory of coherence this appears to be different, one can in principle get access to the reference clock. I do see, however, that there is a certain notion of what may or may not be so difficult to do, and there might be practical considerations that come into play, but in general I don't see how the resource theory of coherence is equally well motivated as the other two mentioned.

In the revised version of the manuscript, it is now stated that:

“In particular, the notion of $\text{\emph{resource distillation}}$, which can be abstractly defined in any resource theory, has a clear operational interpretation in this theory: it is the process in which one combines many noisy clocks, affected by independent noise processes, to obtain less, but more accurate clocks in pure states, which contain more information about time (Note that to maximize the information content of the clock, we should prepare it in a pure state).”

I still don't see a quantitative argument (intuitively I would fully agree though) why it is better to have a pure clock state than a mixed clock state. What is the specific argument, why having fewer clocks with lower entropy is better than many clocks with higher entropy? In particular, the phrase “which contain more information about time” seems odd to me, information quantified by what? Information content quantified by what?

Finally, regarding my remark on the comparison to perpetual motion machines, the manuscript now states:

“In thermodynamics, consideration of the total energy alone is not sufficient to prove the impossibility of perpetual motion machines. Similarly, the impossibility of the hypothetical coherence distillation machine in Fig.1, cannot be shown using the monotonicity of QFI alone. Rather, to prove this one needs to consider a quantity such as the purity of coherence, which distinguishes pure coherence from the total coherence.”

I still don't get what the analogy is supposed to be here, what is written here essentially says: “Statement A alone is not sufficient to prove the impossibility of B. Similarly, the impossibility of C cannot be shown using D alone.” But A and D, as well as B and C do not seem to have a relation

here, so the analogy seems somewhat arbitrary.

In summary, I think the results on distillation of coherence seem correct and I expect them to be of interest to the sufficiently large community of researchers working on this area, with probably more limited interest to researchers outside this circle. Overall, I find the motivation in connection to quantum clocks more tenuous: true, to perform operations beyond those considered one needs a reference clock, but given that one assumes the quantum clocks in question to have been synchronized using this reference clock to begin with, it is unclear to me why one would then only have restricted access to it. Furthermore, it is not quantitatively obvious to me, why one would automatically do better to trade in many mixed clocks for one pure clock. Nevertheless, it seems that readers (in particular, the other referees) are most interested in the resource theory of coherence in itself anyway, where the paper makes some valuable contributions.

Changes asked by the Reviewers:

- 1) I have applied the two minor changes suggested by the reviewer#2.
- 2) Reviewer #3 commented about “the comparison to perpetual motion machines” and the following paragraph in the previous version.

“In thermodynamics, consideration of the total energy alone is not sufficient to prove the impossibility of perpetual motion machines. Similarly, the impossibility of the hypothetical coherence distillation machine in Fig. 1, cannot be shown using the monotonicity of QFI alone. Rather, to prove this one needs to consider a quantity such as the purity of coherence, which distinguishes pure coherence from the total coherence.”

I removed this paragraph and any mention of comparison to perpetual motion machines from the paper.

- 3) Reviewer #3 asked me to clarify the phrase “*which contain more information about time*” in the previous version of the paper, and to explain “why it is better to have a pure clock state than a mixed clock state”.

To address reviewer’s request, I edited the following paragraph at the end of the section on “Distillation of Quantum Clocks” and added a few sentences:

“It is worth emphasizing that the notion of resource distillation, which can be abstractly defined in any resource theory, has a clear operational interpretation in this framework: it is the process in which one combines noisy clocks, affected by independent noise processes, to obtain less, but more accurate clocks in pure states. More precisely, the information content of each output clock about the unknown parameter t , i.e. the current time relative to the standard clock, is greater than the information content of each input clock. Hence, using a distillation protocol, one can increase the efficiency of storage and transmission of quantum clocks.

Intuitively, one expects that to maximize the information content about parameter t , the state of quantum clock should be pure. This intuition is confirmed by the fact that pure states maximize any convex measure of information (about the time parameter t) such as quantum mutual information (Holevo quantity) [19, 20, 27] or quantum Fisher information [27–30]. Similarly, from the point of view of parameter estimation, to minimize the error in the estimation of the time parameter $t \in [0, \tau)$, as quantified by any cost function which is a linear functional of state, such as mean squared error [27, 31], the system should be prepared in a pure state.”